# Multiple Emulsions with Extracts of Cactus Pear Added in A Yogurt: Antioxidant Activity, In Vitro Simulated Digestion and Shelf Life

**DOI:** 10.3390/foods8100429

**Published:** 2019-09-22

**Authors:** Antonio de Jesús Cenobio-Galindo, Gilberto Díaz-Monroy, Gabriela Medina-Pérez, M. Jesús Franco-Fernández, Fanny Emma Ludeña-Urquizo, Rodolfo Vieyra-Alberto, Rafael Germán Campos-Montiel

**Affiliations:** 1Instituto de Ciencias Agropecuarias, Universidad Autónoma del Estado de Hidalgo, Av. Rancho Universitario s/n Km.1, Tulancingo C.P. 43760, Hgo., Mexico; anje_hs@hotmail.com (A.d.J.C.-G.); gilberto.azul.diaz@gmail.com (G.D.-M.); gamepe@yahoo.com (G.M.-P.); mfranco@uaeh.edu.mx (M.J.F.-F.); rodolfo_vieyra@uaeh.edu.mx (R.V.-A.); 2Programa de Doctorado en Desarrollo Científico y Tecnológico para la Sociedad, Centro de Investigación y de Estudios Avanzados del Instituto Politécnico Nacional, Ciudad de México C.P. 07369, Cd. México, Mexico; 3Facultad de Industria Alimentarias, Universidad Nacional Agraria La Molina, Av. la Molina s/n, La Molina, Lima Apdo 12-056, Peru; fludena@lamolina.edu.pe

**Keywords:** color, phenols, flavonoids, betalains

## Abstract

Consumers demand so-called natural in which additive and antioxidant preservatives are from natural origin. Research focuses in using extracts from plants and fruits that are rich in bioactive compounds such as phenolics and betalains, but these are also prone to interact with proteins and are exposed to suffer degradation during storage. In this work, we developed a fortified yogurt with the addition of betalains and polyphenols from cactus pear extract encapsulated in a multiple emulsion (ME) (W_1_/O/W_2_). Different formulations of ME were made with two polymers, gum arabic (GA) and maltodextrin (MD) and with the best formulation of ME four types of yogurt were prepared using different % (*w*/*w*) of ME (0%, 10%, 20% and 30%). Bioactive compounds, antioxidant activity, color and lactic acid bacteria (LAB) were analyzed in the different yogurts over 36 days of shelf life. Furthermore, in vitro simulated digestion was evaluated. The yogurts had significant (*p* < 0.05) differences and the ME protected the bioactive compounds, activity of antioxidants and color. The ME did not affect the viability of LAB during 36 days of storage. The in vitro digestion showed the best bioaccessibilities of antioxidant compounds with the yogurts with ME.

## 1. Introduction

Nowadays people are more interested in consuming functional foods to preserve and/or improve their health. Yogurt is a food produced by the acid coagulation of milk and is very popular [1]. One of the most important advantages of yogurts is it has acid lactic bacteria which improve digestion and therefore health. The Codex Alimentarius [2] established that a minimum of 10^6^ viable lactic acid bacteria/mL are needed to observe beneficial effects in health when consumed in functional foods. 

Several reports indicate that yogurts can be added with some compounds to improve their characteristics [3]. There is currently a growing interest in this type of food containing bioactive compounds of plant origin, which provide benefits as antioxidant activity [4]. Jung et al. [5] incorporated red ginseng extract into a yogurt finding that the antioxidant capacity of the added yogurts was improved. Su et al. [6] elaborated a Hickory milk-based yogurt, finding an important antioxidant activity, effect attributed to the presence of phenolic compounds and small peptides. In addition to some reports with the addition of colorants of natural origin [7,8] to replace artificial colorants that could damage health because its use has been associated with allergies, asthma, hyperactive behavior in children, and it has been associated with an increased risk of cancer [9,10]. 

The bioactive compounds used in foods mainly originate from vegetables [11]. Polyphenols are bioactive compounds that have antioxidant activity [12]. One important source of phenolic compounds (as rutin, ferulic acid, among others) is a cactus pear of the genus *Opuntia*, known as xoconostle, and these compounds have proven to have interesting properties, such as important antioxidant activity [13,14].T there are reports indicating that this fruit is an important source of natural pigments, such as betalains [15]. Unfortunately, these compounds are sensitive to environmental factors, which is why needed to be incorporated into a stable matrix so that they can retain their activity for longer periods [16].

Encapsulation is used to protect compounds and release in a controlled manner [17]. There are different encapsulation methods, one of which is the use of multiple emulsions. There are two types of multiple emulsions: water in oil in water (W_1_/O/W_2_) and oil in water in oil (O_1_/W/O_2_). Multiple emulsions (W_1_/O/W_2_) are capable of protecting to the bioactive compounds in conditions simulated of in vitro digestion [18]. Kreatsouli et al. [19] found than maltodextrin how material of encapsulation for olive mill wastewater phenolic extracts had enhanced the activity antioxidant in human endothelial cell. Pimentel-González [20] evidenced that the encapsulation of polyphenols in multiple emulsions not only protects bioactive compounds, foods can maintain physicochemical characteristics similar to those that have not been added for a longer period. The objective of this study was to develop yogurt with natural pigments and antioxidant compounds from cactus fruit protected through of multiple emulsions to preserve this biological properties in in vitro digestion and increment the shelf life through controlled released of the bioactive compounds.

## 2. Materials and Method

Extracts from cactus pear (*Opuntia oligacantha* C.F. Först) were obtained from the Autonomous University of Hidalgo State. The extracts have phenolics compounds as rutin, ferulic acid, quercetin, 4-hydroxybenzoic acid, apigenin, caffeic acid and kaempferol [14]. A lactic culture, Y1.50B, which is a mixture of Streptococcus thermophiles and *Lactobacillus delbrueckii* subsp. *bulgaricus,* was purchased with a national supplier (Raff S.A, de C.V., and Mexico). Used chemicals and reagents are the following: 2,2′-azino-bis (3-ethylbenzothiazoline-6-sulphonic acid) (ABTS), potassium persulphate, Folin-Ciocalteu reagent, ascorbic acid, 1,1-diphenyl-2-picrylhydrazyl radical (DPPH), pepsin, and quercetin from Sigma-Aldrich (México, Mexico); methanol, acetone, sodium bicarbonate, anhydrous sodium carbonate, hydrochloric acid, ethanol from J.T. Baker S.A; gallic acid and aluminum trichloride from Fermont (Monterrey, N.L., Mexico); Grindsted PGPR 90 (ester of fatty acids of polyglycerol and polyricinoleate) Panodan SDK (esters monoglycerides and diglycerides of tartaric acid diacetyl) from Danisco, México S.A. de C.V.; arabic gum from Frutarom, France; maltodextrin from Grain Processing Corporation Oregon, IA, USA; bile salts OXOID LTD, UK; canola oil From Capullo® (Unilever de México, Tultitlan, Edo. México, Mexico); and pancreatine from Hycel de México S.A. de C.V. México. The medium used was MRS agar (Bioxon- Becton Dickinson México, Edo. México, Mexico).

### 2.1. Multiple Emulsion Formulation

The emulsion was made according to Pimentel-Gonzalez et al. [20] with some modifications. The primary emulsion (W/O) contained 30 mL of cactus pear extract as the internal aqueous phase inside an oil phase of 56 mL canola oil with a mixture of one part hydrophilic emulsifier (Panodan SDK) and four parts hydrophobic emulsifier (Grindsted PGPR 90). The emulsification was carried out with a high shear Ultra-Turrax IKA T25 (digital, Luxembourg, Germany) at 10,000 rpm for 5 min in an ice bath. The double emulsion consisted of 30 mL of primary emulsion with 20 g of polymer and 54 mL of distilled water homogenized at 5 rpm for 5 min in an ice bath. Three formulations were made with different proportions of two polymers; gum arabic (GA) and maltodextrin (MD): GA40%-MD60%; GA50%-MD50%; GA60%-MD40%. In the emulsions were determined droplet diameter, total phenols, flavonoids, betalains and antioxidant activity (ABTS and DPPH). To analyze droplets diameter of the emulsion analysis of the emulsion was taken a sample of one mL and put in slide. It was used optic microscope Olympus BX 45 (Olympus Optical Co. Ltd, Tokio, Japan). It was analyzed 30 droplets using software Image-Pro Plus (version 4.5, Media Cybernetics, Inc., Silver Springs, MD, USA).

### 2.2. Process of Yogurt with the Multiple Emulsions

Whole cow’s milk was used, heated to 45 °C and two percent of milk powder and three percent of sugar were added, mixed until dissolution complete, cool at 42 °C, and it was addition two per cent of lactic culture. The mixture was incubated at 42 °C until has acidity of 70 °D, cool and storage for 12 h at 4 °C, after the refrigeration the suspended fat was withdrawal of superficial and the yogurt was churned to homogenization. Last, the yogurt was mixed with different concentration of the double emulsion. The treatments were following: Yogurt control without ME (Y); Yogurt 10% of ME (Y10); Yogurt 20% of ME (Y20); Yogurt 30% of ME (Y30).

### 2.3. Color

Colorimeter “Hunter Lab” (Minolta, CM508d, Osaka, Japan) was used. The equipment measured darkness and lightness “L”, negative green or positive red value of “a” and negative blue or positive yellow value of “b”. It was taking a sample and it was put in the equipment to obtain the values of coordinates L, a y b [21], three repetitions per treatment were determined.

### 2.4. Total Phenols

To analysis of total phenols was used technique of Folin Ciocalteu described by Ainsworth & Gillespie et al. [22] with some modifications. One mL of sample and mixed with 5 mL of the reactive of Folin Ciocalteu diluted (1:10) after seven minutes was addition 4 mL of sodium carbonate to 7.5%. The samples let stand in the darkness for 120 min. Finally, they were read at 760 nm in spectrophotometer (Jenway 6715, Staffordshire, UK). The results were expressed in mg of gallic acid equivalents for 100 mL (mg GAE/100 mL).

### 2.5. Flavonoids

Total flavonoid content was determined according to Pothitirat et al. [23] with some modifications. A solution of aluminum trichloride (AlCl_3_) (449598 Sigma-Aldrich®, Saint Louis, MO, USA) in 2% methanol (PQ06121 Fermont®, Monterrey, N.L., Mexico) was used. Two mL of the extract was mixed with 2 mL of the methanol solution and placed in the dark to rest for 10 min, after the samples was read at a wavelength of 415 nm. The blank was methanol. The results were expressed in mg of quercetin (1592409 Sigma-Aldrich®, Saint Louis, MO, USA) equivalents for 100 mL of yogurt (mg QE/100 mL).

### 2.6. Betalains

The method of Castellanos-Santiago & Yahia [24] was adapted with some modifications for the determination of betalains. One mL of the sample was mixed with 20 mL of methanol. The mixture stirred and it was read at 483 nm for betaxanthins and 535 nm for betacyanins. The blank was 80 % of methanol. Total content of betalains were expressed in mg/100 mL.

### 2.7. Antioxidant Activity ABTS

The antioxidant activity using the radical 2,2′-azino-bis (3-ethylbenzothiazoline-6-sulphonic acid) (A1888 Sigma-Aldrich®, Saint Louis, MO, USA) was measured by reacting 10 mL of a 7 mM ABTS solution with 10 mL of 2.45 mM (K_2_S_2_O_8_) potassium persulfate (216224 Sigma-Aldrich®, Saint Louis, MO, USA). The mixture was agitated for 16 h in the dark. The cation radical was adjusted with 20% solution of ethanol (100983 Merck^®^, Kenilworth, NJ, USA) to an absorbance of 0.7 ± 0.1 at a 734 nm wavelength; then 200 μL of the extract was added to 2 mL of adjusted ABTS. The reaction occurred for 6 min, after which the absorbance was read [25]. The results were expressed as mg of ascorbic acid equivalents (AAE/100 mL).

### 2.8. Antioxidant Activity DPPH

It was used the methodology described by Reyes-Munguía et al. [26]; the radical 1, 1-diphenyl-2-picrylhydrazyl radical (DPPH) was prepared with a solution 80% of methanol until to obtain a concentration of 6.1 × 10^−5^ M and it was stirred during 2 h in the darkness. It was added 0.5 mL of the sample and 2.5 mL of DPPH and the mixture of both were left reaction for 1 h in darkness and it was read to 515 nm until stabilized. The blank was 80% methanol and the mixture was read at 515 nm. The results were expressed as mg of ascorbic acid equivalents for 100 mL (AAE/100 mL). 

### 2.9. In Vitro Simulated Digestion 

Simulated Intestinal conditions of the yogurt where determined following the method described by Rufián-Henares and Morales [27] with some modifications. There were established two phases: (A) Gastric phase; the extract was diluted (1:5) with distilled water and adjusted to pH 2 by addition of HCl 6 N and 20 mL of gastric liquid (16% of pepsin and 10 % of NaCl in HCL 0.1 M) the mixed was incubated at 37 °C for 2 h in a shaking water bath. When the trial finished two aliquots were taken for the next step and the further analyses; (B) Second phase; the pH of samples from the first trial was adjusted to 7 with sodium bicarbonate (0.5 M) then 1.25 mL of freshly prepared pancreatinin–bile mixture (0.4 g of pancreatinin and 2.5 g of bile salts in 100 mL of 0.1 M NaHCO_3_ (pancreatic fluid). The mixture was incubated at 37 °C for 2 h in a shaking water bath. When the trial finished an aliquot was taken for the further analysis. At the end of the gastrointestinal digestion, the samples was heated in a boiling bath for 4 min in order to inactivate the enzymes and immediately centrifuged at 12,000 rpm for 10 min at 4 °C in a centrifuge Z 36 HK (HERMLE Labortechnik GmbH, Wehingen, Germany) for the analysis of total phenols, flavonoids and antioxidant activity by DPPH and ABTS.

### 2.10. Microbiology Analysis 

The viability of lactic acid bacteria (LAB) was determined through plate culture. The medium used was MRS agar. The incubation was during 24 h at 37 °C and the cells were counts as colony forming unit per mL (CFU/mL), a sterile saline solution was used as a negative control.

### 2.11. Statistics Analysis 

The experimental design was completely randomized. It was used variance analysis with a significance of (*p* < 0.05). When they existed differences significance (*p* < 0.05), it was used medium comparisons technique of Tukey. All the experiments were for triplicate. All data were analyzed using the NCSS 2007 software (Wireframe Graphics, Kaysville, UT, USA).

## 3. Results and Discussion

### 3.1. Multiple Emulsions 

The formulations showed significant differences (*p* < 0.05) in droplets diameter (Figure 1). The formulation GA60%-MD40% had the smaller diameter with droplets of 3.5 µm, this can be due to proportion of arabic gum, which has proteins with emulsifiers properties and stability colloidal against the flocculation [28]. These results were similar to report Di Battista et al. [29] with droplets size 14.7 µm that contain phytosterols used a formulation of gum arabic (75%) and maltodextrin (25%).

The concentration of total phenols, flavonoids and betalains showed significant differences (*p* < 0.05) between the formulated emulsions (Figure 2) during 36 days of storage. The best formulation was GA60%-MD40% keeping 27.3 mg GAE/100 mL of total phenols, 3.3 mg QE/ 100 mL of flavonoids and 1.9 mg/ 100 mL of betalains. These results are higher than reported by Rodríguez-Barahona et al. [30] in phenols in a double emulsion with juice of *Morinda citrifolia*. Comunian et al. [31] showed results similar, a decrement in flavonoids with emulsion of oil during storage and Otálora et al. [32] observed the same behavior in betalains encapsulated of *Opuntia ficus indica.*

The antioxidant activity showed significant differences (*p* < 0.05) between different formulations (Figure 3). The emulsion with higher protection of antioxidant activities (ABTS and DPPH) was GA60%-MD40%, this due to a major concentration of gum arabic in external phase, that help to slow the destabilization of the emulsion [33] in consequence avoid the degradation of the antioxidant compounds. Munguía et al. [34] found the protection of antioxidant compounds from *Opuntia ficus indica* is better with arabig seyal gum.

### 3.2. Characterization of Yogurts with the Multiple Emulsion

#### 3.2.1. Color

The value “L” was increment in proportion of the concentration of the emulsion, due at the maltodextrin, it is a white powder that confer higher value of luminosity [35]. It is observed that the values of L decrease with respect to time.

There are significant differences for the parameter “a” (*p* < 0.05), where it is observed that the treatment that presented higher values was Y30, which means that it tends to red, remembering that thanks to the addition of the multiple emulsions betalains were incorporated that tend to this color.

The parameter “b” was increment in all yogurts during shelf life (Table 1). Comunian et al. [31] reported an increment in the parameter “b” during storage of 30 days with yogurts added with encapsulated of echium oil, phytosterol and sinapic acid.

#### 3.2.2. Total Phenols, Flavonoids and Betalains

All the tested samples from yogurts their quantified amounts of total phenols, flavonoids and betalains tended to decrease over the shelf life (Table 2). The yogurt Y20 had the greatest preservation of bioactive compounds (22.3%, 73.7% and 81.1% of total phenols, flavonoids and betalains, respectively). Muniandy et al. [36] observed similar results: phenols were reduced in refrigerated storage of yogurt with added black tea. The same behavior was found with flavonoids in a yogurt fortified with the grape *Vitis vinífera* by Chouchouli et al. [37]. In the same way, microencapsulate with betalains from purple cactus pear was reduced during storage [38]. 

#### 3.2.3. Antioxidant Activity by ABTS and DPPH

The antioxidant activities (ABTS and DPPH) were similar to bioactive compounds, with significant differences in shelf life and among the yogurts (Table 3). The yogurt Y30 retained more than 70% of its antioxidant activity measured as ABTS (14.9 mg AAE/ 100 mL) over its shelf life. Pimentel et al. [20] reported a decrease in antioxidant activity (ABTS) in cheese fortified with encapsulated phenolic compounds extracted from grape during their maturation. Amirdivani and Baba, [39] found same our results, a trend decreased of the antioxidant activity (DPPH) during storage of 28 days of an herbal yogurt.

#### 3.2.4. LAB Quantification

The LAB had a decrease during storage, how reported Mahrous and Abd-El-Salam [40] in a yogurt fortified with omega-3 and vitamin E. However, all the yogurts had a concentration of lactic acid bacteria (LAB) according to normative [2] with minimum concentration of 1 × 10^6^ CFU/mL in shelf life (Table 3).

#### 3.2.5. In Vitro Simulated Digestion

The yogurt Y30 had the best protection of total phenols (94.6%). In the total phenols showed a lost so much in gastric digestion how intestinal digestion (Table 4). Schulz et al. [41] reported a lost about of 90% in total phenols from jucara, *Euterpe edulis* (Martius) pulp after an in vitro digestion.

The flavonoids presents a different comport in process in vitro digestion. The content of flavonoids was not significant different (*p* > 0.05) in gastric digestion. However, in intestinal digestion stage, there were significant differences (*p* < 0.05) between treatments being again the Y30 the major protection of flavonoids 96.7% (Table 4). The results of protection are higher than reported by Gullon et al. [42] with a loss of 50% with free flavonoids from *Punica granatum* in intestinal digestion due the use of double emulsions.

The Y30 had the better protection of antioxidant activities (ABTS and DPPH) with value of 77.6% and 95.2% respectively (Table 4). Frontela-Saseta et al. [43] reported 65% in antioxidant activity (ABTS) with fruit juices enriched with Pycnogenol at an in vitro digestion. Zudaire et al. [44] found a protection of 31% of antioxidant activity (DPPH) with garlic (*Allium cepa L*.) at an in vitro digestion. The double emulsions improved the protection of bioactive compounds and their antioxidant activities (ABTS and DPPH) in the yogurts.

Food processing can have positive or negative effects on the bioaccessibility of certain compounds. The activity of bioactive compounds is given by multiple factors, such as the release of its matrix, changes in digestion, absorption, metabolism, among others; i.e., these compounds must be bioavailable before acting [16,45]. There are reports regarding the advantages of using double emulsions for encapsulation of bioactive compounds, Jia et al. [46] mention that double emulsions have good stability thanks to their external phase when they are subjected to gastric conditions, either using proteins or polysaccharides, although the latter tend to aggregate in gastric fluids. Ydjedd et al. [47] evaluated the effect of in vitro gastrointestinal digestion of encapsulated phenolic compounds and found that after being subjected to the gastric phase these compounds still continue, which means that their encapsulation protects them against changes in pH and enzymatic activity, contributing to an increase in bioavailability.

Martins et al. [48] incorporated free and encapsulated phenolic compounds of *Rubus ulmifolius* Schott to yogurt and evaluated their activity, finding that these compounds give the matrix antioxidant capacity, finding a better result when the encapsulated extract was used, suggesting that adding encapsulated compounds is an interesting alternative for these products. On the other hand, Caldas-Cueva et al. [49] incorporated pigments from *Opuntia soehrensii* in a yogurt and evaluated their effect, finding that adding the extracts contributed an important source of bioactive compounds, such as betalains, being also used as an effective natural colorant, which suggests a potential use of these pigments to give their color and antioxidant activity to various functional foods, specifically in the dairy industry.

## 4. Conclusions

The formulation AG60%-MD40% had the best stability and major protection of bioactive compounds (total phenols, flavonoids and betalains) and antioxidant activities (ABTS and DPPH). The results show that incorporating bioactive compounds from cactus pear into a multiple emulsions are adequate to prevent the loss of the antioxidant activity and increment the shelf life of yogurt. The multiple emulsions fortified the yogurt with antioxidant activities (ABTS and DPPH) and it give a natural color red, besides it keeps the viability of bacteria acid lactic major of 1 × 10^6^ CFU/mL and a longer shelf life. The total phenols, flavonoids and antioxidant activities (ABTS and DPPH) in the yogurts were protected in the gastric and intestinal digestion by the multiple emulsions. This yogurt with multiple emulsions could improve the health of the consumer due increment of bioaccessibility of bioactive compounds and keep the antioxidant properties after in vitro simulated digestion.

## Figures and Tables

**Figure 1 foods-08-00429-f001:**
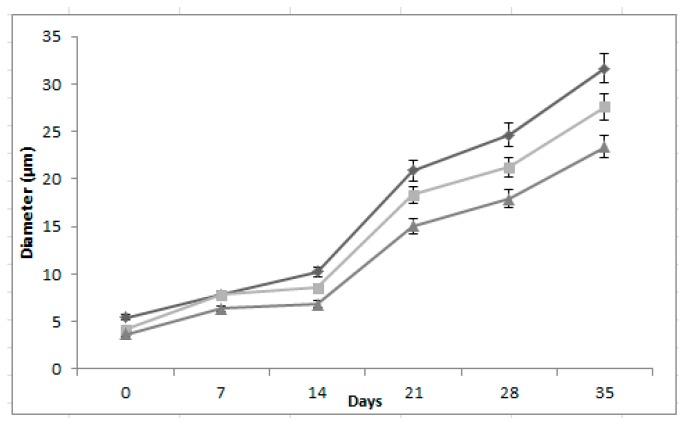
Changes during storage in the diameter of droplets of three multiple emulsions formulated with two polymers, arabic gum (GA) and maltodextrin (MD): GA40%-MD60% (
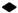
), AG50%-MD50% (
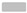
) and AG60%-MD40% (
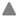
) during storage for 36 days.

**Figure 2 foods-08-00429-f002:**
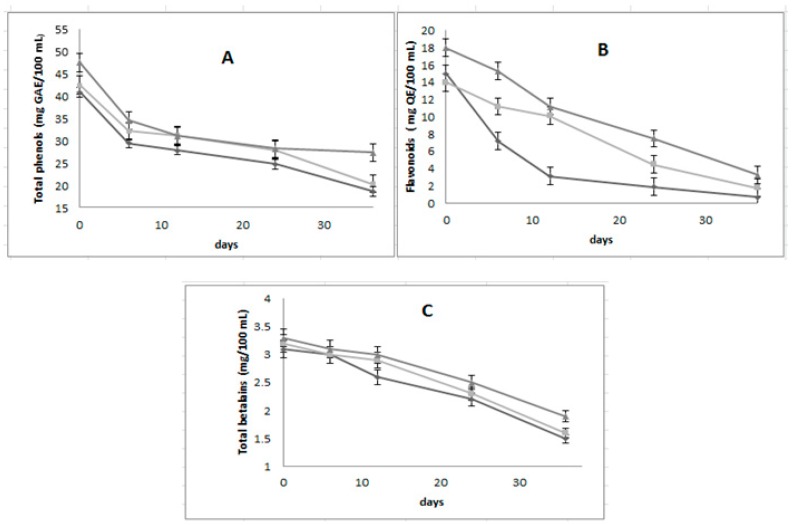
Content of total phenol (**A**), total flavonoids (**B**) and betalains; (**C**) of three multiple emulsions formulated with two polymers, arabic gum (GA) and maltodextrin (MD): AG40%-MD60% (
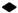
), AG50%-MD50% (
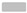
) and AG60%-MD40% (
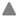
) during 36 days of storage.

**Figure 3 foods-08-00429-f003:**
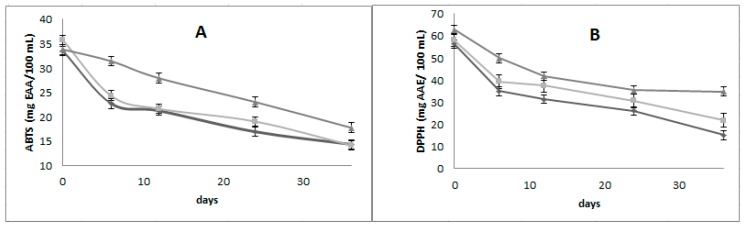
(**A**) Antioxidant activity ABTS and (**B**) Antioxidant activity DPPH of three multiple emulsions formulated with two polymers, arabic gum (GA) and maltodextrin (MD): AG40%-MD60% (
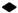
), AG50%-MD50% (
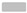
) and AG60%-MD40% (
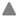
) during 36 days.

**Table 1 foods-08-00429-t001:** Color parameters of L, a y b in different yogurts with different % (*w*/*w*) of multiple emulsions during 36 days of storage.

Days/Yogurts	Y0	Y10	Y20	Y30
		**L**		
0	66.4 ± 0.2 ^cA^	66.5 ± 0.3 ^cA^	67.6 ± 0.3 ^cB^	68.8 ± 0.5 ^cC^
12	64.4 ± 0.4 ^cC^	62.7 ± 0.8 ^bB^	60.3 ± 0.2 ^bA^	60.8 ± 0.2 ^bA^
24	62.1 ± 0.2 ^bB^	61.9 ± 0.4 ^bB^	60.5 ± 0.4 ^bA^	60.0 ± 0.1 ^bA^
36	60.2 ± 0.1 ^aC^	59.9 ± 0.1 ^aC^	58.9 ± 0.1 ^aB^	58.5 ± 0.2 ^aA^
		**a**		
0	−1.2 ± 0.1 ^aA^	1.2 ± 0.05 ^bB^	1.6 ± 0.05 ^cC^	2.1 ± 0.01 ^cD^
12	−1.2 ± 0.05 ^aA^	1.2 ± 0.04 ^bB^	1.5 ± 0.02 ^bcC^	2.0 ± 0.12 ^bcD^
24	−0.2 ± 0.01 ^bA^	1.2 ± 0.03 _bB_	1.4 ± 0.04 ^bC^	1.9 ± 0.09 ^bD^
36	−0.1 ± 0.03 ^bA^	1.1 ± 0.01 ^aB^	1.2 ± 0.03a C	1.7 ± 0.03 ^aD^
		**b**		
0	6.3 ± 0.05 ^aA^	7.3 ± 0.04 ^aB^	7.4 ± 0.02 ^aCD^	7.4 ± 0.02 ^aD^
12	6.5 ± 0.5 ^aA^	8.1 ± 0.1 ^bC^	8.2 ± 0.2 ^bC^	7.7 ± 0.5 ^abBC^
24	7.50.3 ^bA^	9.1 ± 0.1 ^cC^	8.8 ± 0.2 ^cC^	8.25 ± 0.1 ^bB^
36	7.6 ± 0.1 ^bA^	9.31 ± 0.2 ^dD^	8.91 ± 0.1 ^cC^	8.35 ± 0.1 ^bB^

Yogurt control without emulsion (Y), yogurt with 10% of emulsion (Y10), yogurt with 20% of emulsion (Y20) and yogurt with 30% of emulsion (Y30). Different lowercase letters represent a significant difference (*p* < 0.05) within the column (among time) as determined by Tukey´s comparison of averages. Different uppercase letters represent a significant difference (*p* < 0.05) within the row (among yogurts) as determined by Tukey´s comparison of averages.

**Table 2 foods-08-00429-t002:** Total phenols, flavonoids and total betalains in different yogurts added with different % (*w*/*w*) of multiple emulsions during 36 days of storage.

Days/Yogurts	Y0	Y10	Y20	Y30
Total phenols (mg GAE/100 mL)				
0	ND	17.9 ± 0.1 ^dA^	38.5 ± 0.4 ^dB^	50.7 ± 0.1 ^cC^
12	ND	12.7 ± 0.2 ^cA^	20.1 ± 0.1 ^cB^	23.1 ± 0.1 ^bC^
24	ND	7.7 ± 0.1 ^bA^	10.3 ± 0.5 ^bB^	14.5 ± 0.1 ^aC^
36	ND	6.4 ± 0.2 ^aA^	8.6 ± 0.3 ^aB^	9.9 ± 0.2 ^aC^
Flavonoids (mg QE/100 mL)				
0	ND	19.9 ± 0.1 ^dA^	25.1 ± 0.5 ^cB^	23.8 ± 0.1 ^cC^
12	ND	18.2 ± 0.2 ^cA^	21.1 ± 0.5 ^bB^	28.3 ± 0.3 ^bC^
24	ND	16.8 ± 0.5 ^bA^	19.1 ± 0.3 ^aB^	26.1 ± 0.7 ^aC^
36	ND	15.9 ± 0.2 ^aA^	18.5 ± 0.5 ^aB^	25.1 ± 0.5 ^aC^
Total betalains (mg/100 mL)				
0	ND	0.52 ± 0.01 ^cA^	1.1 ± 0.04 ^cB^	1.4 ± 0.06 ^bC^
12	ND	0.5 ± 0.05 ^bcA^	1.0 ± 0.03 ^bB^	1.3 ± 0.04 ^bC^
24	ND	0.5 ± 0.1 ^bcA^	0.9 ± 0.01 ^aB^	1.1 ± 0.1 ^aC^
36	ND	0.46 ± 0.01 ^aA^	0.9 ± 0.02 ^aB^	1.0 ± 0.02 ^aC^

Yogurt control without emulsion (Y), yogurt with 10% of emulsion (Y10), yogurt with 20% of emulsion (Y20) and yogurt with 30% of emulsion (Y30). Different lowercase letters represent a significant difference (*p* < 0.05) within the column (among time) as determined by Tukey´s comparison of averages. Different uppercase letters represent a significant difference (*p* < 0.05) within the row (among yogurts) as determined by Tukey´s comparison of averages. ND = no detected.

**Table 3 foods-08-00429-t003:** Antioxidant activities (ABTS and DPPH) and lactic acid bacteria in different yogurts added with different % (*w*/*w*) of multiple emulsions during 36 days of storage.

Days/Yogurts	Y0	Y10	Y20	Y30
ABTS (mg AAE/100 mL)				
0	15.8 ± 1.6 ^bA^	15.8 ± 1.6 ^bA^	16.3 ± 1.4 ^cA^	21.0 ± 1.3 ^cB^
12	1.8 ± 0.1 ^aA^	14.0 ± 0.5 ^bB^	14.7 ± 1.4 ^cB^	17.9 ± 0.3 ^bC^
24	1.6 ± 0.1 ^aA^	10.1 ± 0.8 ^bA^	11.8 ± 0.6 ^bB^	16.8 ± 0.3 ^aC^
36	1.6 ± 0.01 ^aA^	10.1 ± 1.4 ^aB^	9.1 ± 0.3 ^aB^	14.9 ± 0.3 ^aC^
DPPH (mg AAE/100 mL)				
0	2.1 ± 0.02 ^aA^	51.2 ± 0.2 ^dA^	52.7 ± 0.8 ^dB^	52.9 ± 0.3 ^cC^
12	2.1 ± 0.04 ^aA^	18.2 ± 0.9 ^bB^	29.8 ± 0.2 ^cD^	26.1 ± 0.7 ^bC^
24	2.0 ± 0.03 ^aA^	12.5 ± 01 ^aB^	17.7 ± 0.3 ^bC^	11.0 ± 1.9 ^aB^
36	1.8 ± 0.1 ^bA^	11.1 ± 0.2 ^aC^	9.9 ± 0.3 ^aB^	9.3 ± 0.3 ^aB^
Lactic acid bacteria (LAB) (CFU × 10 ^6^ mL)				
0	31.7 ± 0.6 ^cA^	31.3 ± 0.5 ^cA^	31.6 ± 0.7 ^cA^	31.3 ± 0.5 ^cA^
12	28.3 ± 1.5 ^bcC^	25.1 ± 0.1 ^bB^	24.6 ± 0.5 ^bB^	21.3 ± 0.6 ^bA^
24	25.3 ± 0.5 ^bc^	16.3 ± 1.1 ^aB^	13.4 ± 0.6 ^aA^	12.3 ± 0.5 ^aA^
36	18.3 ± 2.4 ^aC^	14.7 ± 1.5 ^aB^	12.6 ± 1.7 ^aA^	11.9 ± 0.7 ^aA^

Yogurt control without emulsion (Y), yogurt with 10% of emulsion (Y10), yogurt with 20% of emulsion (Y20) and yogurt with 30% of emulsion (Y30). Different lowercase letters represent a significant difference (*p* <0.05) within the column (among time) as determined by Tukey´s comparison of averages. Different uppercase letters represent a significant difference (*p* < 0.05) within the row (among yogurts) as determined by Tukey´s comparison of averages. ND = no detected.

**Table 4 foods-08-00429-t004:** Total Phenols, flavonoids and antioxidant activities (ABTS and DPPH) in different yogurts added with different % (*w*/*w*) of multiple emulsions in vitro digestion.

Yogurts	Total Phenols (mg GAE/ 100 mL)			Flavonoids (mg QE/100 mL)		
	Without treatment	Gastric digestion	Intestinal digestion	Without treatment	Gastric digestion	Intestinal digestion
Y0	ND	ND	ND	ND	ND	ND
Y10	19.9 ± 0.3 ^cA^	17.5 ± 0.1 ^bA^	16.4 ± 0.2 ^aA^	17.9 ± 0.3 ^bA^	18.2 ± 0.4 ^bA^	17.4 ± 0.2 ^aA^
Y20	39.6 ± 0.2 ^cB^	38.2 ± 0.1 ^bB^	37.1 ± 0.3 ^aB^	28.5 ± 0.5 ^bB^	28.9 ± 0.3 ^bB^	27.1 ± 0.4 ^aB^
Y30	51.9 ± 0.3 ^cC^	50.1 ± 0.2 ^bC^	49.2 ± 0.2 ^aC^	30.5 ± 0.2 ^bC^	31.1 ± 0.2 ^bC^	29.5 ± 0.3 ^aC^
	**ABTS (mg AAE/100 mL)**			**DPPH (mg AAE/100 mL)**		
	Without treatment	Gastric digestion	Intestinal digestion	Without treatment	Gastric digestion	Intestinal digestion
Y0	1.8 ± 0.01 ^A^	ND	ND	2.0 ± 0.02 ^A^	ND	ND
Y10	15.76 ± 0.03 ^cA^	13.3 ± 0.03 ^bA^	10.7 ± 0.15 ^aA^	51.56 ± 0.01 ^cB^	49.5 ± 0.14 ^bA^	47.25 ± 0.06 ^aA^
Y20	16.3 ± 0.09 ^cB^	14.3 ± 0.14 ^bB^	12.1 ± 0.02 ^aB^	52.06 ± 0.02 ^cC^	50.11 ± 0.11 ^bB^	48.33 ± 0.11 ^aB^
Y30	21.0 ± 0.11 ^cC^	19.2 ± 0.07 ^bC^	16.3 ± 0.05 ^aC^	52.84 ± 0.05 ^cD^	51.68 ± 0.16 ^bC^	49.18 ± 0.08 ^aC^

Yogurt control without emulsion (Y), yogurt with 10% of emulsion (Y10), yogurt with 20% of emulsion (Y20) and yogurt with 30% of emulsion (Y30). Different lowercase letters represent a significant difference (*p* < 0.05) within the row (among time) as determined by Tukey´s comparison of averages. Different uppercase letters represent a significant difference (*p* < 0.05) within the column (among yogurts) as determined by Tukey´s comparison of averages.

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
