# Peer review of "Multiple Emulsions with Extracts of Cactus Pear Added in A Yogurt: Antioxidant Activity, In Vitro Simulated Digestion and Shelf Life"

_foods, 2019, doi:10.3390/foods8100429_

Round 1
Reviewer 1 Report
This is an interesting paper regarding the beneficial properties of a fortified yogurt. However, it is not written properly, whereas the conclusions and the potential applications of the findings are not well presented. Hereby there are some major comments than need to be addressed by the authors.
Comments
The introduction is too short. The authors should expand it by adding extra relevant information. For example, they could refer to the biological activities of polyphenols, gum arabic and maltodextrin in more detail. Please consult a relative paper (https://doi.org/10.3390/antiox8080280).
The objective of the study is not clear. The authors should explain the advantages of the fortified yogurt they developed and its putative applications.
The resolution of the figures is poor.
The authors should also add one or two paragraphs in the discussion explaining their findings because the way they do it, their conclusions are vague to the readership.
The English language needs improvement throughout the manuscript.
Reviewer 2 Report
“FORTIFIED YOGUR WITH MULTIPLE EMULSIONS: EFFECTS OF THE INCORPORATION BIOACTIVE COMPOUNDS OF CACTUS PEAR ON SHELF LIFE AND in vitro DIGESTION” for Food journal. The subject of the paper is interesting. This the manuscript is written in an easy-to-follow, pedagogical style. The work is well cited throughout. I have only a few concern related to the manuscript.
The title needs to be revised and simplified. Did the author analyze the extracts from cactus pear (Opuntia oligacantha C.F. Först) thorough HPLC and mass spectrophotometer to find out the composition (bioactive compounds in it)? The author has to include the sensory test and the antimicrobial test to prove the quality of the modified yogurt as compared to the controlled yogurt. Some of the sentences are repeated in the in vitro digestion “ The yogurt ME30 had the best protection of total phenols (94.6%). The yogurt ME30 had the best 221 protection of total phenols (94.6%).”
Round 2
Reviewer 1 Report
The authors have successfully addressed my comments.